# Comparison of Pulmonary Function and Inflammation in Children/Adolescents with New-Onset Asthma with Different Adiposity Statuses

**DOI:** 10.3390/nu14142968

**Published:** 2022-07-20

**Authors:** Xiaolan Ying, Jie Lin, Shuhua Yuan, Chunhong Pan, Wenfang Dong, Jing Zhang, Lei Zhang, Jilei Lin, Yong Yin, Jinhong Wu

**Affiliations:** Department of Respiratory Medicine, Shanghai Children’s Medical Center, School of Medicine, Shanghai Jiao Tong University, No. 1678 Dongfang Road, Pudong, Shanghai 200127, China; lanlancherish@163.com (X.Y.); linjie@scmc.com (J.L.); shuhuaysh@hotmail.com (S.Y.); pch73@126.com (C.P.); dwf20220103@163.com (W.D.); zhangjing@scmc.com.cn (J.Z.); scmc_zhang@163.com (L.Z.); jilei_lin@163.com (J.L.)

**Keywords:** children, adolescents, asthma, obesity, body mass index (BMI), pulmonary function, inflammation, atopy

## Abstract

(1) Background: The relationship between obesity and asthma is still uncertain. This study aimed to investigate the effect of overweight/obesity on the pulmonary function of patients with new-onset pediatric asthma and explore the possible causative factors related to concomitant obesity and asthma. (2) Methods: Patients aged 5 to 17 years old with newly diagnosed mild to moderate asthma were recruited from June 2018 to May 2019, from a respiratory clinic in Shanghai, China. Participants were categorized into three groups: normal weight, overweight, and obese asthma. A family history of atopy and patients’ personal allergic diseases were recorded. Pulmonary function, fractional exhaled nitric oxide (FeNO), eosinophils, serum-specific immunoglobulins E (sIgE), serum total IgE (tIgE), and serum inflammatory biomarkers (adiponectin, leptin, Type 1 helper T, and Type 2 helper T cytokines) were tested in all participants. (3) Results: A total of 407 asthma patients (197 normal weight, 92 overweight, and 118 obese) were enrolled. There was a reduction in forced expiratory volume in the first second (FEV1)/forced vital capacity (FVC), FEV1/FVC%, and FEF25–75% in the overweight/obese groups. No difference was found between the study groups in the main allergy characteristics. Leptin levels were higher while adiponectin was lower in asthmatics with obesity. Higher levels of IL-16 were found in overweight/obese asthmatic individuals than in normal-weight individuals. (4) Conclusions: Obesity may have an effect on impaired pulmonary function. While atopic inflammation plays an important role in the onset of asthma, nonatopic inflammation (including leptin and adiponectin) increases the severity of asthma in overweight/obese patients. The significance of different levels of IL-16 between groups needs to be further studied.

## 1. Introduction

The prevalence of both asthma and overweight/obesity has increased worldwide in recent years. Epidemiological studies [1,2,3,4,5,6,7] have shown that obesity increases the risk of asthma onset and the severity, and appears to worsen asthma control. Children with obesity and asthma appear to have increased susceptibility to the effects of environmental pollutants [8]. Asthma symptoms improved as a result of weight loss strategies [9]. This phenomenon has resulted in an increasing number of prescribed medications and decreased quality of life for patients. Although there is growing evidence of an “obese asthmatic” phenotype [10,11], its specific characteristics are still not completely understood.

Type 2 helper T (Th2)-mediated inflammation has been widely considered to be the main driver of allergic asthma [12]. More Th2 cells induce increased eosinophils in peripheral blood and eosinophil-mediated airway damage through chemotaxis to the airways. Interleukin (IL)-4, which is associated with the Th2 response, acts as one of the key cytokines mediating this progression. Persons with atopy have an exaggerated production of sIgE antibodies, which can be tested in serum [13]. However, the role of atopy or allergic airway inflammation in the “obese asthmatic” phenotype is currently unclear.

Previous studies on the impact of obesity on asthma have focused on inflammatory factors. Obesity is a chronic systemic inflammatory disease characterized by the accumulation of multiple adipose-tissue deposits throughout the body and an increase in circulating cytokines. Adipokines (leptin, adiponectin, etc.) are secreted by adipocyte tissue from adipocytes [14]. It is currently believed that adiponectin and leptin act independent of Th2 cells in anti-inflammatory and pro-inflammatory activities, respectively [15,16]. Systemic nonallergic inflammation, with Type 1 helper T (Th1)-predominant immune patterns such as elevated levels of IL-6, tumor necrosis factor (TNF)-α, and interferon (IFN)-γ, is another potential underlying mechanism of the obese asthma phenotype [17]. The increase in the concentration of cytokines may be the main pathophysiological basis for obesity’s ability to exacerbate asthmatic airway inflammation [18].

In addition, different metabolic factors, such as hemoglobin A1c, are supposed to be associated with the obese asthma phenotype in children [19]. Additionally, excessive adipose-tissue accumulation in obesity leads to limited mechanical movement of the thorax, decreased lung compliance, and reduced peripheral-airway diameter, which leads to asthma [20,21]. 

Our study was conducted to elucidate the factors initiating and affecting the disease process by comparing the differences in atopic and nonatopic inflammation between normal pediatric asthmatic patients and overweight/obese patients. Understanding the effect of obesity on asthma is important for the implementation of prevention and early therapeutic strategies for the obese asthma phenotype.

## 2. Materials and Methods

### 2.1. Study Design

This prospective cross-sectional study was conducted from June 2018 to May 2019. The study population was recruited from the outpatient clinic of the Respiratory Department, Shanghai Children’s Medical Center Affiliated to Shanghai Jiao Tong University School of Medicine, and patients were enrolled in the order of consultation. 

### 2.2. Subjects

The inclusion criteria were as follows: children/adolescents aged between 5 and 17 years; who were diagnosed for the first time with asthma (mild to moderate) according to the guidelines of the global initiative for asthma (GINA) [22]; who were able to cooperate and undergo respiratory function tests; and who underwent all the related tests. Criteria for exclusion from the study included: systemic steroid therapy in the last 4 weeks; chronic diseases except for obesity or atopy (eczema, atopic dermatitis, rhinitis, food allergy, etc.); combined with acute infection; and exposure to active or passive smoking. Demographic variables, including age, sex, ethnicity, and living area, were collected.

### 2.3. Anthropometry

Weight and height were measured by trained observers with standardized equipment, and the results were recorded. Body mass index (BMI) was calculated as (weight; kg)/[(height; m)^2^]. According to the World Health Organization (WHO) 2007 growth reference [23], participants with an age- and sex-specific BMI between the 5th and 85th percentile were categorized as “normal weight”, those between the 85th and 95th percentile were categorized as “overweight”, and those at or above the 95th percentile were categorized as “obese”.

### 2.4. Clinical Assessment

A family history of atopy and patients’ personal histories of other allergic diseases were recorded. A family history of atopy was defined as the presence of a first-degree relative with asthma, rhinitis, or atopic dermatitis. The clinical diagnosis of atopic diseases (eczema, atopic dermatitis, rhinitis, atopic conjunctivitis, food allergy, etc.) was performed by a pediatric allergist. Whether there were any previous physician-diagnosed atopic diseases was reported by parents. 

Complete blood counts (including eosinophil counts) in peripheral blood were obtained. According to previous studies, an eosinophil ratio was ≥4% was evaluated as eosinophilia. 

### 2.5. Pulmonary Function Tests

Forced expiratory volume in the first second (FEV1), forced vital capacity (FVC), and forced expiratory flow between 25% and 75% of vital capacity (FEF25–75) are expressed as a percentage of the predicted value (% pred.). FEV1/FVC is expressed as the best value and a percentage of the predicted value (% pred.).

Spirometry was performed for all the patients by a single pulmonologist on the day of the physical examination. The JAEGER MasterScreen Pneumo spirometer (Jaeger, Germany) was used to evaluate pulmonary ventilation function. All procedure rules were in accordance with the American Thoracic Society (ATS) and the European Respiratory Society (ERS) recommendations [24], and the Respiratory Branch of Chinese Pediatric Society of Chinese Medical Association Editorial Board of Chinese [25,26]. 

### 2.6. Fractionated Exhaled Nitric Oxide (FeNO)

FeNO was measured (NIOX MINO, Solna, Sweden) as a sensitive marker of eosinophilic airway inflammation and is expressed as parts per billion (ppb). The standardized procedures of FeNO measurement were based on the ATS/ERS recommendations and the Respiratory Branch of the Chinese Pediatric Society [27,28]. We set 20 ppb as the cutoff value of high/low airway inflammation [29].

### 2.7. Serum-Specific Immunoglobulins E (sIgE) and Serum Total IgE Test

Baseline sIgE tests for common aeroallergens and food allergens were then performed by the AllergyScreen test (Mediwiss Analytic GmbH, Moers, Germany). sIgE levels ≥ 0.70 IU/mL were considered to be positive [30]. Serum total IgE levels were measured by nephelometry (Beckman Coulter Immage 800, Clare, Ireland) in all patients.

### 2.8. Adiponectin, Leptin and Cytokine Measurement

Leptin (RayBiotech, Peachtree Corners, GA, USA) was quantified by ELISA according to the manufacturer’s directions. The Magnetic Luminex Assay Human Premixed Multi-Analyte Kit (R&D Systems, Minneapolis, MN, USA) was applied to measure the concentration of the following chemokines and cytokines (according to the manufacturer’s instructions): adiponectin; and the cytokines interferon (IFN)-γ, IL-4, IL-16; and tumor necrosis factor (TNF)-α. All data were collected by the Luminex-100 system (Luminex, Austin, TX, USA).

### 2.9. Statistical Analysis

Statistical analysis was performed using the SPSS software version 26.0 (SPSS, Inc., Chicago, IL, USA). Analytical methods (Kolmogorov—Smirnov/Shapiro—Wilk test) were used to determine whether the variables were normally distributed. Descriptive analyses are presented as the mean and standard deviation (SD) for normally distributed variables, and the median and interquartile range (IQR) for non-normally distributed variables. Categorical variables are presented as numbers and percentages. The comparison of averages of continuous normally distributed variables between the three groups was performed using ANOVA with LSD post hoc tests. A t test was used to compare the averages of continuous normally distributed variables between two groups. A Kruskal—Wallis test was conducted to compare the differences in non-normally distributed variables. The chi square test or Fisher’s exact test was used for the comparison of categorical variables. Bonferroni’s correction was used for a posteriori comparisons. A significance level of 0.05 was considered for all tests performed. For multiple comparisons, we considered *p* values below a Bonferroni-corrected α = 0.01(α = 0.05/5 outcomes) to indicate significance. 

## 3. Results

### 3.1. Study Population

A total of 407 patients (197 normal weight, 92 overweight, and 118 obese) were enrolled in this study. Table 1 illustrates the basic characteristics of the study population and the intergroup comparisons. Males (67.3% of the whole participants) accounted for a higher percentage in each group. The average age of all participants was 7.27 years, and no significant difference between groups was found (*p* = 0.21). Participants enrolled in this study were all of Han ethnicity, lived in Shanghai, and did not have exposure to secondhand smoke.

### 3.2. Pulmonary Function Analysis among Different Weight Categories

Table 2 demonstrates respiratory function measurements and the interdifferences among subjects. No significant difference was found in FVC% or FEV1% among the three groups. Conversely, the FEV1/FVC ratio, FEV1/FVC (%pred), and FEF25–75% were significantly different among the three study groups. Compared with the normal weight or the overweight group, these three indicators were significantly decreased in the obese group, which indicates a vulnerably reduced pulmonary function. However, comparison between the normal weight and overweight groups showed no difference in any of these spirometric variables. 

Statistical differences in the FEV1/FVC ratio and FEV1/FVC (%pred) were still found between the normal weight group and the combined overweight/obese group (Appendix A).

### 3.3. Association of BMI with Atopic Characteristics

The mechanism of the onset of the obese asthma phenotype has not been clearly clarified. We further investigated the possible different causes of new-onset asthma among children/adolescents in different BMI categories by analyzing clinical and laboratory characteristics. The main allergic characteristics are presented in Table 3, including sociodemographic characteristics, clinical features, and immunologic indicators. A total of 46.4% of patients with asthma had a family history of atopy, and no significant difference was found between the three groups. The majority of asthma patients (90.7%) had other allergic diseases (rhinitis, eczema, atopic dermatitis, conjunctivitis, food allergy, etc.). A total of 76.4% of the study population had allergen sensitivity with a positive sIgE test. We also compared the total IgE level, blood eosinophil cellularity, interleukin (IL)-4, and FeNO. There was no significant difference between the groups in any of those measures.

A statistical analysis of the comparison between the normal-weight group and the combined overweight/obese group is also presented (Appendix A).

### 3.4. Comparison of Nonatopic Inflammation in the Study Groups

New-onset asthma in obese individuals is thought to have a noneosinophilic immunopathological mechanism. We further compared several nonatopic inflammation indicators.

The comparison of leptin and adiponectin between groups is shown in Table 4. Leptin and adiponectin levels were significantly different between the three weight categories. For asthmatic patients with normal weight, leptin levels were relatively low and adiponectin levels were higher. We also measured the IFN-γ and TNF-α. The concentrations in obese asthmatic children were similar to those of normal weight, either for IFN-γ or TNF-α.

Table 4 also presents the intergroup comparison of IL-16 levels. Asthmatic children with normal weight had a lower level of IL-16 than the other groups.

We further compared the combined overweight/obese group with the normal-weight group. Statistically significant differences could still be found in leptin, adiponectin, and IL-16 between the two groups (Appendix A).

## 4. Discussion

### 4.1. Effects of Obesity on Pulmonary Function in Asthmatic Children/Adolescents

The relationship between asthma and obesity in children/adolescents is still controversial. Since impaired pulmonary function could be responsible for the symptom profile and a reduction in functional status, the effects of obesity on pulmonary function were assessed. One recent meta-analysis concluded that regardless of asthma status, overweight/obesity impaired lung function in children of different age groups [31].

The FEV1/FVC ratio is commonly used as a measure of the degree of airflow obstruction and severity of asthma. The present study found reductions in the FEV1/FVC ratio and FEV1/FVC (%pred) in overweight or obese participants, which confirmed the obstructive pattern of pulmonary function in overweight/obese asthmatics. Consistent with our results, decrements in the FEV1/FVC were noted with BMI increases in adolescents with asthma [32]. Similar results have also been reported in another study, in which FEV1/FVC in the sitting position was significantly lower in overweight/obese asthmatic children than in normal-weight children [33].

The baseline FEF25–75% was significantly reduced in obese patients showing a lower peripheral airflow compared to that in those with normal weight or overweight. Similarly, previous studies also showed a lower FEF25–75% in overweight or obese asthmatic children than in normal-weight children [31,34].

Our study added evidence to the prior literature demonstrating certain pulmonary-function deficits in overweight/obese pediatric asthmatic patients. Due to the cross-sectional nature of the study, the relationship between obesity and pulmonary function cannot be used to deduce causality. Whether these findings help to explain the impact of obesity on the severity of new-onset asthma requires further study.

### 4.2. Th2-Cell-Mediated Eosinophilic Inflammation Might Play a Role in Obese Asthma Patients

Th2-cell-mediated eosinophilic airway inflammation has been demonstrated as the main mechanism in normal-weight children with asthma [35]. 

We found no difference in the prevalence of atopy among different BMI percentile groups. This observation suggested a preponderant atopic component in the onset of asthma, regardless of obesity status. Except for the IL-4 level, other features related to atopy (including the presence of other allergic diseases, family history of atopy, the positivity of sIgE, eosinophils, etc.) seemed to be more apparent in overweight patients, but no significant difference was found among groups. The divergence may be due to the relatively small sample size. These results were in accordance with those of other studies [34,36]. The similar atopic inflammatory profile might further suggest that the standard treatment with inhaled corticosteroids could be effective in obese asthmatic patients. However, findings by other researchers indicated that obese asthmatic individuals did not exhibit allergic inflammation [37] and commonly had a noneosinophilic inflammation [38,39]. One meta-analysis indicated that systemic immune responses are nonatopic among obese asthmatics and differ from atopic inflammation among normal-weight asthmatic individuals [40]. 

The discordant findings might have resulted from differences in study design, the control of confounders and/or difference in the study cohort characteristics, e.g., age, sex [41], ethnicity [41], early or late-onset asthma [42], body fat distribution [43], the sample size, etc.

### 4.3. Possible Mechanism of Nonatopic Inflammation Related to Leptin and Adiponectin in Obese Asthmatic Patients

The lack of atopy difference between the study groups suggests that the lower pulmonary function in obese asthmatics may be affected by obesity-mediated inflammation. Based on the evidence that obesity precedes asthma, we speculate that a possible mechanism could be that long-term adiposity status plays a role in reduced pulmonary function.

Adiponectin is negatively correlated with BMI [15,44]. However, in our study, overweight asthmatic individuals had the lowest level among the groups. This might be because the adiponectin levels varied over a wide range and were distributed non-normally, and our sample size was relatively small. After combining the overweight and obese individuals into one group, the combined group still showed a significantly lower level of adiponectin. Adiponectin is supposed to take part in anti-inflammatory activities by downregulating pro-inflammatory cytokines, e.g., TNF-α and IL-6 [15], and upregulating anti-inflammatory cytokines, such as IL-10 [15,45]. Adiponectin deficiency is thought to increase allergic inflammation and pulmonary vascular remodeling in a chronic asthma model [46]. 

Obese patients had a higher level of leptin, which is consistent with other studies [15,44]. This result suggests the possible mechanism by which leptin might play a role in the onset of overweight/obese asthma. Leptin has proinflammatory activities and affects both innate and adaptive immune responses [47]. Several population-based studies in prepubertal children described the association between the presence of asthma and serum leptin levels [48,49]. Comparing serum leptin in overweight/obese asthmatic individuals with that in controls with a similar BMI would further confirm the effect of leptin on asthma. One such study [50] in peripubertal participants showed leptin levels twice as high as those in the control participants. Furthermore, leptin expression has been found in the lung and has a great impact on lung development [51].

In addition, the obese asthma phenotype is associated with Th1-predominant inflammation, which gives rise to lower pulmonary function [39]. However, in our study population, no significant difference in cytokines associated with the Th1 response (IFN-γ and TNF-α) was found among the three asthmatic groups. We speculate that adipose-tissue inflammation might lead to an obese asthma phenotype in various ways. In-depth mechanistic studies on obese asthma are required to clarify the contributions of leptin, adiponectin, nonallergic systemic Th1 immune patterns, and Th2 inflammation, respectively. 

### 4.4. The Role of IL-16 in Inflammation

The biological function of IL-16 and its implications in allergic diseases remain to be determined. Our study demonstrated that IL-16 levels were higher in overweight/obese asthmatic individuals than in those with normal weight, which is consistent with another study conducted with 79 adolescents [52]. Similarly, an animal study [53] showed that eosinophil IL-16 content correlated directly with donor BMI. No association between any of the other eosinophil cytokines (IL-1β, IL-1α, and IL-6) and IL-16 levels was found. This finding suggested that eosinophil-focused directions considering the relationships between adiposity, eosinophils, and IL-16 need to be further studied.

IL-16 has a well-known impact on the increased chemotactic activity of CD4+ cells in asthmatic patients [54]. The study conducted by Afifi S.S. et al. [55] concluded that measuring IL-16 levels may be used to evaluate the severity of airway inflammation. While IL-16 can be immunostimulatory, both in vitro and in vivo research has also found that it has immunosuppressive effects on a Th2-dominated allergic response [56,57,58]. This might reveal that the elevated IL-16 level in asthmatic patients is a result of inflammation rather than a causative factor. At the cellular level, IL-16 was identified as synthesized by preadipocytes, although it did not exhibit a response to the proinflammatory mediator IL-1β [59].

We speculate that higher levels of IL-16 resulted from more severe inflammation in the overweight/obese groups. The immunosuppressive effect of IL-16 might be associated with weight. However, the exact mechanism remains to be explored. Only a limited number of previous evaluations considered the specific impact of IL-16 and its correlation with weight. An improved understanding of the relationship between BMI and IL-16 is worth further study.

### 4.5. Highlights and Limitations

The present study investigated the interaction between asthma and obesity in a multifaceted manner. All participants were new-onset patients with mild to moderate disease who were diagnosed in our clinic and had not undergone systemic steroid therapy in the past 4 weeks. Thus, the carefully selected patients had good comparability. Our results confirmed the effect of atopy and nonatopic inflammation in asthmatic patients, which assisted in directing asthma therapy. Due to the scarcity of studies in children/adolescents, these results added evidence of the relationship between BMI and the severity of new-onset asthma. Therefore, our study contributes to a better knowledge of the obese asthma phenotype and could inform the direction of more comprehensive studies.

There are some limitations in this study. This was a single-center study that lasted for one year, and participants were not recruited at random. The sample size may have limited the identification of differences among groups, and important covariates (such as sex, atopic status, timing of obesity onset, and asthma exacerbation) were not taken into account. The cross-sectional design hinders the inference of causality. Additionally, normal-weight, nonasthmatic children/adolescents and obese, nonasthmatic children/adolescents could have been investigated as control groups. In addition, a comparison between before and after treatment remains to be investigated.

## 5. Conclusions

Obesity may have an effect on impaired pulmonary function. The present study investigated the possible causative factors of asthma in overweight/obese children/adolescents in several aspects. While atopic inflammation plays important roles in the onset of asthma, nonatopic inflammation (including leptin and adiponectin) aggravates the severity of the overweight/obese asthma phenotype. We further found that IL-16, a cytokine which has immunosuppressive effects, is more prominent in the obese asthma phenotype. The prevention and therapy of obese asthma may require emphasizing atopy and adiposity.

## Figures and Tables

**Table 1 nutrients-14-02968-t001:** Demographic and anthropometric characteristics of the study patients and comparison of subgroups.

	Total *n* = 407	Normal Weight *n* = 197	Overweight *n* = 92	Obese *n* = 118	*p*
Sex					
Male *n* (%)	274 (67.3)	111 (56.3)	60 (65.2)	103 (87.3)	**<0.001**
Female *n* (%)	133 (32.75)	86 (43.7)	32 (34.8)	15 (12.7)	
Age (years)					
Mean	7.72	7.49	7.84	7.94	0.21
SD	2.22	2.08	2.36	2.27	
BMI (kg/m^2^)					
Mean	18.22	15.57	18.91	22.02	**<0.001**
SD	3.51	1.61	1.88	2.98	

Bold indicates *p* < 0.05.

**Table 2 nutrients-14-02968-t002:** Comparison of spirometric variables among the study groups.

	Total Mean (SD)	Normal Weight Mean (SD)	Overweight Mean (SD)	Obese Mean (SD)	*p*
FVC (%pred)	101.03 ± 15.82	101.62 ± 15.98	100.80 ± 12.75	100.21 ± 17.71	0.74
FEV1 (%pred)	96.36 ± 17.99	97.87 ± 18.64	96.88 ± 16.95	93.41 ± 17.46	0.10
FEV1/FVC (%)	80.97 ± 8.76	82.08 ± 8.51 ^a^	81.39 ± 8.95 ^a^	78.78 ± 8.69 ^b^	**0.005** ^#^
FEV1/FVC (%pred)	95.00 ± 10.14	96.09 ± 9.87 ^a^	95.62 ± 10.28 ^a^	92.69 ± 10.20 ^b^	**0.01**
FEF25–75 (%pred)	69.13 ± 26.10	71.14 ± 26.52 ^a^	71.28 ± 26.62 ^a^	64.02 ± 24.45 ^b^	**0.04**

Bold indicates *p* < 0.05. Different letters (a,b) indicate significant differences between the two study groups (*p* < 0.05). ^#^ indicates if below Bonferroni-adjusted *p* < 0.01 (0.05/5 outcomes).

**Table 3 nutrients-14-02968-t003:** Sociodemographic, clinical, and immunologic features of the three groups.

	Total	Normal Weight	Overweight	Obese	*p*
Family history of atopy					
No *n* (%)	218 (53.6)	111 (56.3)	39 (42.4)	68 (57.6)	0.05
Yes *n* (%)	189 (46.4)	86 (43.7)	53 (57.6)	50 (42.4)	
Other allergic disease					
No *n* (%)	38 (9.3)	22 (11.2)	4 (4.3)	12 (10.2)	0.17
Yes *n* (%)	369 (90.7)	175 (88.8)	88 (95.7)	106 (89.8)	
Single allergic disease	155 (38.1)	74 (37.6)	43 (46.7)	38 (32.2)	
Multiple allergic diseases *	214 (52.6)	101 (51.3)	45 (48.9)	68 (57.6)	
Positive sIgE test					
No *n* (%)	96 (23.6)	53 (26.9)	14 (15.2)	29 (24.6)	0.09
Yes *n* (%)	311 (76.4)	144 (73.1)	78 (84.8)	89 (75.4)	
Total IgE (IU/mL)					
Median	295.00	320.00	351.50	246.00	0.09
IQR	135.00–647.00	166.50–649.00	130.50–724.50	97.15–604.75	
Eosinophils (×10^9^/L)					
Median	0.35	0.36	0.37	0.33	0.98
IQR	0.13–0.64	0.09–0.68	0.14–0.57	0.17–0.60	
Eosinophils% ≥ 4% *n* (%)	205 (50.4)	95 (48.2)	50 (54.3)	60 (50.8)	0.62
FeNO ≥ 20ppb *n* (%)	213 (52.3)	107 (54.3)	53 (57.6)	53 (44.9)	0.14
IL-4 (pg/mL)					
Median	100.05	102.64	101.71	95.91	0.31
IQR	88.95–100.59	90.52–110.59	87.36–115.60	86.66–108.99	

* a combination of 2 or more allergic diseases. IQR: Interquartile range; eosinophils%: percentage of eosinophils in blood cells.

**Table 4 nutrients-14-02968-t004:** Comparison of leptin, adiponectin, and serum cytokine levels between different groups.

	Total Median (IQR)	Normal Weight Median (IQR)	Overweight Median (IQR)	Obese Median (IQR)	*p*
Leptin (ng/mL)	3.18 (1.16–6.54)	1.05 (0.72–2.70) ^a^	3.45 (1.57–6.92) ^b^	6.28 (3.34–13.47) ^b^	0.002 ^#^
Adiponectin (μg/mL)	68.39 (53.00–96.95)	86.42 (65.40–126.82) ^a^	60.22 (44.18–85.85) ^b^	67.57 (53.99–94.68) ^a,b^	0.04
TNF-α (pg/mL)	30.07 (26.45–36.06)	28.75 (26.84–31.38)	32.13 (26.69–39.51)	30.03 (24.56–35.63)	0.45
IFN-γ (pg/mL)	7.07 (3.98–9.03)	6.54 (3.76–8.62) ^a^	7.79 (5.54–9.60) ^b^	6.81(3.38–9.01) ^a,b^	0.62
IL-16 (pg/mL)	316.20 (233.67–445.66)	237.04 (211.00–319.45)	362.30 (305.64–509.74)	307.51(225.95–471.53)	0.03

Bold indicates *p* < 0.05. Different letters (a,b) indicate significant differences between the two study groups (*p* < 0.05). ^#^ indicates if below Bonferroni-adjusted *p* < 0.01 (0.05/5 outcomes).

## Data Availability

The datasets are available from the corresponding author upon reasonable request.

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
