# Peer review of "Comparison of Pulmonary Function and Inflammation in Children/Adolescents with New-Onset Asthma with Different Adiposity Statuses"

_nutrients, 2022, doi:10.3390/nu14142968_

Round 1
Reviewer 1 Report
This is an interesting study by Ying et al. Regarding changes in pulmonary function end inflammatory markers in children and adolescents with newly diagnosed asthma and differences in adiposity status. This manuscript would benefit significantly by being edited by a professional editing service designed to present the work in scientific English; the language barrier comes through very strongly and, at times, makes it difficult to read the work comprehensively.
Major issues
1. Although differences between groups are presented in some tables (e.g., table 2), the methodology for Post hoc analysis is not described. Furthermore, the manuscript could benefit from such analysis in the rest of the results.
2. Table 1 should include ranges for the values provided. Other baseline characteristics, including previous medications (if any), should also be included.
3. Data and statistics presented in tables 2, 3, and 4 should be corrected for multiple comparisons and preferably include 95% confidence intervals.
4. Several of the results presented in tables 3 and 4 regarding overweight children seem to not agree with the overall trend. For example, overweight children have lower adiponectin and higher TNFa, IFNg, and IL-16 than obese children. Please comment on this. Would the combined overweight/obese group analysis provide a more straightforward message?
5. The discussion should be extensively rewritten to reflect the paper's findings and their relevance to the literature. Currently, the discussion is too extensive and feels like a review.
Minor issues
1. Lines 218-219. "Our study added evidence to prior literatures demonstrating that pulmonary function decreased across increasing BMI percentile categories in pediatric asthmatics". Please correct this since BMI percentile categories are not presented in the analysis of the results.
2. The authors should also discuss the role of adiposity in the natural history of asthma. One would expect that the pro-inflammatory state of obese children will affect asthma severity during the course of the disease more than it will affect it at the initial stages of asthma diagnosis.
3. The research sample was one of convenience, and this is a significant limitation that should be reported.
Author Response
Major issues
- Although differences between groups are presented in some tables (e.g., table 2), the methodology for Post hoc analysis is not described. Furthermore, the manuscript could benefit from such analysis in the rest of the results.
Reply:
The pulmonary function (table 2) between groups was analyzed with ANOVA and posteriori comparisons were calculated by LSD method.
Mann-Whitney U test was used to compare nonnormally distributed variables (leptin, adiponectin, serum cytokines levels) between the different groups; Pearson Chi-square test or Fisher’s exact test was used for comparison of categorical variables; Bonferroni's correction was given at posteriori comparisons.
- Table 1 should include ranges for the values provided. Other baseline characteristics, including previous medications (if any), should also be included.
Reply:
We added it in the results, “Participants enrolled in this study were all of Han ethnicity, lived in Shanghai, and did not have exposure to secondhand smoke.”.
As it was mentioned in the Method, patients with chronic diseases (except other allergic diseases) were excluded in this study. And all the patients had not been treated with systemic steroid in the last 4 weeks. As they were newly diagnosed asthmatic patients, none had standardized treatment for asthma ever before. Thus, their previous medications were not described particularly.
- Data and statistics presented in tables 2, 3, and 4 should be corrected for multiple comparisons and preferably include 95% confidence intervals.
Reply:
We mentioned it in the Method part.
“For multiple comparisons, we considered p values below a Bonferroni-corrected α = 0.01(α = 0.05/5 outcomes) to indicate significance.”
Data in tables 3, and 4 have been corrected for multiple comparisons.
There’s no statistic difference between each group in tables 2. Thus, multiple comparisons were not presented in tables 2.
- Several of the results presented in tables 3 and 4 regarding overweight children seem to not agree with the overall trend. For example, overweight children have lower adiponectin and higher TNFa, IFNg, and IL-16 than obese children. Please comment on this. Would the combined overweight/obese group analysis provide a more straightforward message?
Reply:
Although several results in table 3 shown that children in overweight group had the highest/lowest level, no statistical difference can be found among the groups. The comparison of the combined overweight/obese group with the normal weight group was provided in Supplementary material, Table 1.
The results of leptin, adiponectin, serum cytokines levels seem to not agree with the overall trend. This might be because the level of these indicators varied over a wide range and the sample size is relatively small.
We further compared the combined overweight/obese group with the normal weight group. Statistically significant differences could still be found in leptin, adiponectin and IL-16 between the two groups (Supplementary material, Table 3).
- The discussion should be extensively rewritten to reflect the paper's findings and their relevance to the literature. Currently, the discussion is too extensive and feels like a review.
Reply:
The discussion has been extensively rewritten to focus on the main findings and key points of this study.
Minor issues
- Lines 218-219. "Our study added evidence to prior literatures demonstrating that pulmonary function decreased across increasing BMI percentile categories in pediatric asthmatics". Please correct this since BMI percentile categories are not presented in the analysis of the results.
Reply:
It has been corrected into “Our study added evidence to prior literatures demonstrating certain pulmonary function deficits in overweight/obese pediatric asthmatics.”
- The authors should also discuss the role of adiposity in the natural history of asthma. One would expect that the pro-inflammatory state of obese children will affect asthma severity during the course of the disease more than it will affect it at the initial stages of asthma diagnosis.
Reply:
The present study was mainly about the effect of obesity on the severity (expressed by spirometric variables) of patients with new-onset asthma.
We added it in the Discussion. “One recent meta-analysis concluded that regardless of asthma status, overweight/obesity impaired lung function in children of different age groups.”
- The research sample was one of convenience, and this is a significant limitation that should be reported.
Reply:
We mentioned it in the Limitation part:
This is a single-center study lasted for one year and participants were not recruited at random.
Reviewer 2 Report
Please add this comparation rlation ship with this papers on your discussion, Please change dicussion with the new references and Clarificy more graph number 1 And conclussion please separete points and try answer more clarificy about hipotesis
1 Cite Share Effects of body weight and posture on pulmonary functions in asthmatic children. Afr Health Sci. 2020 Dec;20(4):1777-1784. doi: 10.4314/ahs.v20i4.31.PMID: 34394239 2 Cite Share Pediatric metabolic outcome comparisons based on a spectrum of obesity and asthmatic symptoms. J Asthma. 2019 Apr;56(4):388-394. doi: 10.1080/02770903.2018.1463377. Epub 2018 May 3.PMID: 29676936 3 Cite Share A Behavioral Family Intervention for Children with Overweight and Asthma. Clin Pract Pediatr Psychol. 2018 Sep;6(3):259-269. doi: 10.1037/cpp0000237. Epub 2018 May 24.PMID: 30416909 4 Cite Share Obesity may enhance the adverse effects of NO2 exposure in urban schools on asthma symptoms in children. J Allergy Clin Immunol. 2020 Oct;146(4):813-820.e2. doi: 10.1016/j.jaci.2020.03.003. Epub 2020 Mar 18.PMID: 32197971 5 Cite Share Comparison of hypothesis- and data-driven asthma phenotypes in NHANES 2007-2012: the importance of comprehensive data availability. Clin Transl Allergy. 2019 Mar 13;9:17. doi: 10.1186/s13601-019-0258-7. eCollection 2019.PMID: 30918624
Author Response
Comments and Suggestions for Authors
Please add this comparation relationship with this papers on your discussion, Please change dicussion with the new references and Clarificy more graph number 1
And conclussion please separete points and try answer more clarificy about hypotesis
Reply:
Thanks for your suggestion. We’ve rewritten the discussion. Comparation relationship and new references have been added.
We tried to clarify the hypothesis in conclusion.